



# Ability of an Australian reanalysis dataset to characterise sub-daily precipitation

Suwash Chandra Acharya[1], Rory Nathan[1], Quan J Wang[1], Chun-Hsu Su[1,2], Nathan Eizenberg[2]

[1]Department of Infrastructure Engineering, the University of Melbourne, Australia
[2]Bureau of Meteorology, Melbourne, Australia

*Correspondence to*: Suwash Chandra Acharya (suwasha@student.unimelb.edu.au)

**Abstract.** The high spatio-temporal variability of precipitation is often difficult to characterise due to limited measurements. The available low-resolution global reanalysis datasets inadequately represent the spatio-temporal variability of precipitation relevant to catchment hydrology. The Bureau of Meteorology Atmospheric high-resolution Regional Reanalysis for Australia

(BARRA) provides a high-resolution atmospheric reanalysis dataset across the Australasian region. For hydrometeorological applications, however, it is essential to properly evaluate the sub-daily precipitation from this reanalysis. In this regard, this paper evaluates the sub-daily precipitation from BARRA for a period of 6 years (2010-2015) over Australia against point observations and blended radar products. We utilise a range of existing and bespoke metrics for evaluation at point and spatial scales. We examine bias in quantile estimates and spatial displacement of sub-daily rainfall at a point scale. At a spatial scale,

we use the Fractions Skill Score as a spatial evaluation metric. The results show that the performance of BARRA precipitation depends on spatial location with poorer performance in tropical relative to temperate regions. A possible spatial displacement during large rainfall is also found at point locations. This displacement, evaluated by comparing the distribution of rainfall within a day, could be quantified by considering the neighbourhood grids. On spatial evaluation, hourly precipitation from BARRA are found to be skilful at a spatial scale of less than 100km (150km) for a threshold of 75% quantile (90% quantile)

at most of the locations. The performance across all the metrics improves significantly at time resolutions higher than 3 h. Our evaluations illustrate that the BARRA precipitation, despite discernible spatial displacements, serves as a useful dataset for Australia, especially at sub-daily resolutions. Users of BARRA are recommended to properly account for possible spatio-temporal displacement errors, especially for applications where the spatial and temporal characteristics of rainfall are deemed very important.

## 1 Introduction

Precipitation is highly variable across both space and time, especially at spatial and temporal scales relevant to catchment hydrology (Michaelides et al., 2009). An understanding of the spatio-temporal pattern of precipitation is vital for many scientific and operational applications such as hydro-climatic modelling and the forecasting of floods (Golding et al., 2016; Kucera et al., 2013; Paschalis et al., 2014). This understanding relies on access to high-resolution precipitation datasets.



However, the availability of fine-scale precipitation products (e.g. spatial resolution less than around 0.25° at hourly time
      scale) are limited. The general sources of precipitation data are gauge measurements, ground-based radars, satellites and
      atmospheric reanalysis models (Michaelides et al., 2009). Gauge measurements are hindered by sparse measurement and
      uneven density of gauge network, whereas the coverage of ground-based radars is limited. Global reanalysis datasets (e.g.
      NCEP-CFSR, Saha et al., 2010; ERA-Interim, Dee et al., 2011; JRA-55, Kobayashi et al., 2015) and satellite products
(e.g.TMPA, Huffman et al., 2007; IMERG, Huffman et al., 2018)  provide a continuous and consistent estimate at varying
      spatial (0.05° to 2.5°) and temporal resolution (hourly to daily).

      An atmospheric reanalysis merges observations and models to provide a four-dimensional earth system data at a regular spatial
      and temporal resolution over a long time period, often years and decades (Parker, 2016). The variables in the reanalysis (such
as precipitation, cloud and soil moisture) are related to one another through modelled physical relations and with the analysed
      observations. By undertaking modelling over a limited area, a regional atmospheric reanalysis can provide precipitation
      estimates at finer spatial and temporal scale than a global reanalysis. It can incorporate more observations at a finer scale to
      better constrain the evolution of a higher-resolution model, and thus can account for the effects of mountains, coastlines, and
      mesoscale atmospheric circulations in greater detail. Such analyses can thereby provide precipitation estimates with greater
spatial relevance to local fine-scale studies than coarser-scale models. BARRA (Bureau of Meteorology atmospheric high-
      resolution regional reanalysis for Australia) is one such high-resolution (12 km) regional reanalysis (BARRA-R). It is driven
      by the global ~79 km ERA-Interim reanalysis and provides estimates over the Australasian region from 1990 to 2018 (Jakob
      et al., 2017; Su et al., 2019).

BARRA-R (referred to here as simply BARRA) provides hourly precipitation estimates at 12 km horizontal resolution (Jakob
      et al., 2017; Su et al., 2019). Rainfall observations are not assimilated in this reanalysis and precipitation is estimated by model
      physics and parameterisation. Evaluating its suitability for scientific studies and developing use cases can be facilitated by
      identifying its strengths and limitations through a quantitative assessment. An initial assessment of daily precipitation from
      BARRA showed that it was able to reproduce the precipitation statistics and large precipitation at point (gauged locations) and
grid scale of 5 km (Acharya et al., 2019; Su et al., 2019). However, precipitation data at sub-daily temporal resolutions are
      essential to support the application of flood modelling (Chiaravalloti et al., 2018) and analysis of precipitation extremes in
      convective systems. Therefore, it is useful to assess the performance of BARRA at sub-daily resolution along with its ability
      to represent the spatial structure of rainfall events.

The performance of BARRA at a daily scale was summarised using continuous metrics and categorical metrics at both point
      and grid scales (Acharya et al., 2019). However, at sub-daily scales, such one-on-one evaluations can be misleading for several
      reasons (Jermey and Renshaw, 2016). First, sub-daily precipitation is dominated by zero values and has a highly skewed
      distribution, which makes it difficult to interpret the correlation statistics. In addition, these metrics doubly penalise a modelled





rainfall field that is displaced in space or time: once as a missed observation, and again as a false alarm. This situation is

popularly known as "double-penalty problem" (Rossa et al., 2008), and to mitigate this issue it is necessary to adopt an evaluation approach which considers likely displacements in space and time (Gilleland et al., 2009; Jermey and Renshaw, 2016; Thiemig et al., 2012). For example, when evaluating heavy rainfall events at point scale, Thiemig et al. (2012) adopted an error metric that explicitly considers possible time-lags in the gridded dataset. Similarly, Jermey and Renshaw (2016) considered temporal displacement while evaluating the precipitation by grouping the events into seven synthetic categories

with varying temporal shift and bias structure. Besides these methods which look at temporal displacement, a possible shift across space can be considered when evaluating high-resolution precipitation datasets. However, such evaluations may be limited by the availability of suitable gridded observation data for comparison, and it is thus preferable to undertake assessments using multiple evaluation techniques (Jermey and Renshaw, 2016).

The metrics used to undertake the evaluation need to match the prospective use of the dataset. For example, it may be sufficient for some urban and stormwater design problems to only evaluate the non-conditional frequency distribution of precipitation at a point. However, for most hydrological modelling purposes, it is necessary to evaluate the representation of spatio-temporal characteristics of rainfall events over the catchment of interest. The spatial and temporal resolution of the prospective applications also play a critical role in selection of the performance measures, so it is useful to assess the efficacy of the

reanalysis over a range of different temporal and spatial accumulations.

The objective of this paper is to present an evaluation of sub-daily BARRA precipitation over temporal and spatial resolutions that are relevant to catchment hydrology applications. The evaluation is undertaken using ground measurements and radar-based rainfall observations across Australia over a six-year period spanning 2010-2015. Following the assessment of BARRA

at daily scale (Acharya et al., 2019), this study further explores BARRA precipitation at different time accumulations ranging from hourly to daily. In addition, we compare the temporal distribution of gauged rainfalls with neighbouring grids, specifically for large rainfall events. We also employ a spatial evaluation metric (Roberts and Lean, 2008) to evaluate the ability of the reanalysis dataset to represent the spatial distribution of rainfalls against a gauge-corrected radar dataset. Overall, our aim is to assess the ability of BARRA to capture the behaviour of sub-daily precipitation at the catchment scale, particularly for large

events. To this end, we use a combination of existing and bespoke metrics to evaluate performance over different temporal and spatial accumulations in different climatic zones.

## 2 Study area and data sources

The primary sources of reference data used to evaluate the performance of BARRA data were derived from pluviometer gauges and radar-based products. Ground-based observations are not assimilated in BARRA and thus these data serve as an

independent dataset for evaluation. The study areas for the analysis of pluviometer observations encompass three of the broad



climatic zones across Australia, namely tropical, temperate and arid. The radar-based evaluation is undertaken at four city-centred regions that are located in these three zones (Brisbane, Darwin, Melbourne and Sydney).

Hourly rainfall observations from pluviometer gauges were obtained from the Australian Bureau of Meteorology. Observations from a total of 441 gauges were selected, covering a common period of record between 2010 and 2015 (Figure 1). The observed rainfall is used to approximate precipitation as most precipitation in Australia is in the form of rainfall.

The blended radar precipitation dataset was also obtained from the Australian Bureau of Meteorology for regions surrounding the four city centres (see Figure 1). The spatial resolution of the radar data is ~1.5 km, and it is only available from 2014. The radar fields have been blended with the observed rainfall using conditional merging (Sinclair and Pegram, 2005). The approach is in principle close to a copula-based approach, which exhibits less bias and yields a smaller error metric compared to non-parametric radar rainfall estimation (Hasan et al., 2016). The blended radar estimates are still likely to be erroneous and biased due to reasons that include: errors in reflectivity measurement (e.g., radar beam overshoot, terrain blocking, clutter), inaccurate radar reflectivity and rain rate relationship, tendency of the radar to underestimate rainfall at distance, and quality control algorithms rejecting gauge data used for bias adjustment (Chumchean et al., 2006; Seo et al., 2010). Further, there is a fundamental difference in representativity between radar measurements and modelled precipitation, where radar infers precipitations at height over a cubic kilometre in size. However, we follow prior studies (e.g. Mittermaier et al., 2013; Roberts and Lean, 2008) and use radar rainfall to compare spatial rainfall at sub-daily time steps assuming that the spatial distribution of rainfall is preserved. In addition, we apply quantile-based thresholds in order to remove the potential bias in the daily rainfall totals, although we note the strategy remains limited by the fact that the bias is spatially varying.

BARRA utilises the Unified Model (UM, Davies et al., 2005) and its 4D variational data assimilation system (4D Var). BARRA extends spatially over 65.0° to 196.9° east, -65.0° to 19.4° north at a spatial resolution of 0.11° (approximately 12 km) and with 70 levels up to 80 km into the atmosphere. The model includes a comprehensive set of parametrisations, including a modified boundary layer scheme, mixed phase cloud microphysics, a mass flux convection scheme, and a radiation scheme. The model parameterisation in BARRA is mainly inherited from the UKMO Global Atmosphere (GA) 6.0 configurations as described in Walters et al. (2017). Observations from land surface stations, ships, buoys, aircrafts, radiosondes, and satellites are assimilated in BARRA, conducted 4 times a day with a 6-hour analysis window centred at time $t_0 = 0, 6, 12$ and 18 UTC. Surface and satellite rainfall observations are not assimilated, and the precipitation fields are determined by the modelled dynamics. In particular, they are estimated from the 12-hour (h) forecast runs of the UM from $t_0 - 3h$, using the microphysics scheme based on Wilson and Ballard (1999) and the mass flux convective parameterisation scheme of Gregory and Rowntree (1990). The former describes the atmospheric processes that transfer water between the various states of water to remove moisture resolved on the grid scale. At 12 km horizontal resolution, BARRA requires the convection scheme to model sub-grid scale convection using an ensemble of cumulus clouds as a single entraining-detraining plume (Clark et al., 2016). The

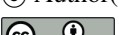



scheme prevents unstable growth of cloud structures on the grid and explicit vertical circulations, and can only predict an area-average rainfall instead of a spectrum of rainfall rates. Further details on parameterisation and assimilation schemes in BARRA are provided by Su et al. (2019).

## 3 Methodology

Hourly BARRA precipitation estimates at 12 km resolution are evaluated at varying spatial and temporal scales. A range of
point-to-grid and grid-to-grid evaluations are undertaken using gauged rainfall and blended radar estimates over various temporal and spatial scales.

### 3.1 Frequency distribution of rainfall

A point-to-grid analysis is undertaken to evaluate the sub-daily frequency distribution of BARRA precipitation. At each gauge location as shown in Figure 1, corresponding BARRA estimates are obtained using the nearest neighbour interpolation. The
basis of comparison requires some thought as the fraction of zero values in sub-daily rainfall data is high (the 95% quantiles at hourly temporal resolution are zero). This issue could be addressed by selecting suitable conditional thresholds, though different thresholds would need to be adopted for each temporal accumulation. For example, a threshold of 0.1mm/h could be adopted for hourly data and 1 mm/day for daily. However, the slight problem with this approach is that any trends in the performance metric with temporal accumulation will be confounded by the somewhat arbitrary choice of thresholds for
intermediate temporal accumulations (3h, 6h, and 12h). Accordingly, we derived quantiles for sub-daily rainfalls that occurred only on the rain days for each dataset individually, where, a "rain day" is defined based on a threshold of 1 mm/day (Ebert et al., 2007).

In investigating the frequency of rainfall, we compute the various quantiles (80, 90, 95, 99%) at different time accumulations
(1 h, 3 h, 6 h, 12 h and 24 h). We then estimate percentage bias using *(m-o)/o * 100*, where *m* and *o* denote the reanalysis and observed precipitation corresponding to the selected quantiles. This evaluation is further stratified across three broad climatic zones (arid, tropical, and temperate) as defined by the Köppen-Geiger classification (Peel et al., 2007; Figure 1). It is worth noting that comparisons of point (gauged) and areal (gridded) rainfall are generally biased as the average precipitation over a grid cell is lower than rainfall recorded at a particular point. It would also be expected that sub-daily point rainfalls are more
variable than those averaged over a grid cell area.

### 3.2 Spatio-temporal distribution

A direct comparison of precipitation, especially in higher resolution datasets, is difficult as the conventional metrics are not able to penalise intensity and location errors in a desirable manner (Rossa et al., 2008). Performance metrics which compare





point observations with model estimates averaged over a grid cell are heavily influenced by errors in the spatial pattern and
location of rainfall events, even if the average rainfall depths over the local domain are the same. This is particularly the case
with high-resolution precipitation datasets, and with the analysis of sub-daily periods.

To mitigate this problem in a gauge-based evaluation, we adopt an approach that allows for possible timing and displacement
differences in rainfall occurrence. The approach is similar to the neighbourhood (or "fuzzy") approach used with single
observation-neighbourhood forecasts (Ebert, 2008), in which gridded observations were evaluated against gridded forecasts.
This method accounts for situations where an event defined on the basis of gauge measurements may miss the nearby grid cell
resulting in spatial error, and/or appear non-coincident at the nearby grid cell signifying a temporal error. For the former case,
an evaluation that considers neighbouring grid cells can account for spatial errors. For cases involving a time displacement, a
moving storm may appear at a neighbouring grid cell at the time-step under consideration, approaching towards the nearby
grid cell.

To account for these different types of errors, we employ an analysis that explicitly considers the occurrence and timing of
rainfalls at neighbouring grid cells. This evaluation is undertaken by selecting large rainfall events (greater than 10mm/day)
in the gauge dataset. The cumulative fraction of rainfall over the day is computed for the gauge and the nearest neighbouring
grid (Eq. 1). The squared difference in the fraction of cumulative rainfall occurring in each hour between the gauge and
reanalysis rainfalls is then averaged to provide an error score (Eq. 2).

$$F_h = \sum_{i=1}^{h} f_i, \qquad 1 \le h \le 24 \tag{1}$$

where, $f_i$ is the fraction of daily rainfall occurring at i$^{\text{th}}$ hour.

$$error\ score = \frac{1}{23} \sum_{h=1}^{23} \left( F_{h,model} - F_{h,gauge} \right)^2 \tag{2}$$

Note that 23 is used in the denominator as $F_{24,\ model} = F_{24,\ gauge} = 1$ and are not included in the computation of the error score.
The temporal error is considered by using cumulative precipitation at a daily scale, which penalises large temporal errors more
than small ones. Similarly, to account for possible spatial displacement, we analyse the temporal distribution of precipitation
by searching over neighbourhood space to find the best performing grid cell. The error score is computed for both the nearest
grid and the neighbourhood grid cells equidistant from the nearest grid. The minimum error score is selected and then averaged
across all rainfall events at a location.

The average error score varies between 0 and 1. A score of 1 represents the worse possible situation in which rainfall occurs
in the first hour in one dataset and the last hour in the other. Conversely, a score of 0 indicates a perfect match between
observations and model estimates. Scores between 0 and 1 indicate differing degrees of temporal error, where for example, a





score of 0.33 indicates that rainfall occurs in either the first or last hour in one data set and is distributed uniformly throughout the day in the other.

### 3.3 Fractions skill score (FSS)

Finally, a spatial evaluation of reanalysis precipitation against blended radar estimates is undertaken. For each spatial domain, the largest 25 rainfall events to have occurred over the 3-year period are selected based on domain-averaged daily precipitation. Radar precipitation, which is available as 30-minute accumulations, is aggregated to hourly to match the temporal resolution of BARRA. Similarly, a common spatial resolution is adopted. The precipitation from BARRA (~12km) is re-gridded to the resolution of the radar grid (~1.5km) using area-weighted re-gridding, which means that single BARRA precipitation estimates

are distributed over 8×8 radar grid cells.

Different metrics have been developed for undertaking spatially variable evaluations (Ebert, 2008, 2009). This study adopts the FSS from Roberts and Lean (2008), as it measures the variation of skill across increasing spatial scales and hence indicates the minimum spatial scale at which the model is skilful.


The FSS metric is based on the likelihood that rainfall over a given threshold occurs somewhere within the neighbouring window of grid cells. A common threshold rainfall rate is selected for both the observed and modelled grid cells. An increasing window of size $n \times n$ centred on a particular grid is selected (which yields $N$ windows over the whole domain). For each window $i$, a fraction of grid points exceeding the threshold in observed rainfall ($p_{o,i}$) and modelled rainfall ($p_{m,i}$) are computed.

Then, the FSS is calculated as:

$$FSS = 1 - \frac{\frac{1}{N}\sum_{i=1}^{N}(p_{m,i} - p_{o,i})^2}{\frac{1}{N}\sum_{i=1}^{N}p_{m,i}^2 + \frac{1}{N}\sum_{i=1}^{N}p_{o,i}^2} \qquad (3)$$

The FSS score varies between 0 and 1. A score of 0 represents a complete mismatch between two rainfall fields and a score of 1 represents a perfect match. Usually, FSS is computed for varying size of spatial windows and results are plotted as a function

of window size. A random score (FSS$_{random}$) is the FSS that would be achieved, on average, by a random field with the same fraction of observed events ($p_o$) over the domain. A benchmark score, a target or 'uniform' skill (FSS$_{uniform}$), is given to a uniform field with a probability of occurrence equal to $p_o$ at each grid cell (Roberts and Lean, 2008). FSS$_{uniform}$ is expressed as 0.5+$p_o$/2, which is halfway between a perfect score (1) and a random score (FSS$_{random}$ = $p_o$). This FSS$_{uniform}$ can be approximated to 0.5 when $p_o$ is small, as in the case of larger precipitation thresholds.






The thresholds used to calculate FSS are based on rainfall quantiles in observed and reanalysis data which are derived for each time step. A rainfall intensity greater than 0.2 mm/h is used as a threshold to define a rainy grid cell for which quantile-based thresholds are computed. This ensures that the model and observed rainfall fields have an identical fraction of rain events for each threshold value. The application of quantile based thresholds aims to remove the impact of any bias in rainfall amount

and focus solely on spatial accuracy of modelled precipitation (Mittermaier et al., 2013; Roberts and Lean, 2008; Skok and Roberts, 2018).

## 4 Results

### 4.1 Frequency distribution of rainfall

Figure 2 illustrates the spatial distribution of quantile bias and its summary across climatic zones. Overall, the spatial

distribution of bias across all time accumulations and quantiles exhibit a similar pattern. The biases in the northern region are higher and spatially more variable than those in the southern region. The variation in bias across quantiles is the highest for hourly rainfall, especially in arid and tropical climatic zones. For the 80% quantile of hourly rainfalls, all stations in the arid and tropical zones exhibit a higher positive bias than those in the temperate zone. This difference is partly due to the tendency of BARRA to overestimate light rain events. Despite considering wet days only in comparing quantiles, hourly rainfall at 80%

quantile for arid and tropical region dominated by small precipitation amount which results in a high positive bias. At higher quantiles, the BARRA estimates are all lower than the gauged point rainfalls. This negative bias is largest in the tropical zone, followed by the arid zone. Although the nature of the high hourly bias varies with location, a step reduction in bias is observed when the accumulation time periods increase from 1 to 3 and 6 hours. This is partly due to reduced inherent bias arising from the adoption of longer temporal accumulations which reduces the potential for differences in timing between observations and

model estimates.

The spatial distribution of biases associated with the different quantiles exhibits a similar pattern for all temporal accumulations considered. At shorter accumulations (up to 3 hours), the biases change from positive to negative and gradually increase in magnitude with increasing quantile. At higher accumulations (6, 12 and 24 hours), the bias is negative but decreases in

magnitude with increasing quantiles. At higher quantiles (95% and 99%), the bias is the least at the temperate zone. The BARRA estimates are under-predicted at all time accumulations, but this improves with increasing temporal accumulations.

### 4.2 Spatio-temporal patterns

Figure 3 shows how the minimum error score changes with varying neighbourhood grid size. It is seen that the error score decreases significantly when a neighbourhood of 3×3 grid cells (about 35×35 km) is considered instead of a single nearest





neighbour only. The error score continues to decrease slightly as the size of the neighbourhood increases, though the adoption of a larger neighbourhood increases the likelihood that rainfall events unrelated to the gauge observations are being considered. The results show that the temporal distribution of the BARRA precipitation estimates within a wet day are representative yet displaced in terms of location. The error scores are all less than 0.33, which indicates that the temporal distribution of sub-daily BARRA precipitation is on average superior to a uniform distribution of estimates derived by simply disaggregating

daily rainfalls over a day.

### 4.3 Fractions Skill Score (FSS)

Figure 4 shows the hourly precipitation rates corresponding to different quantile thresholds in the Brisbane, Darwin, Melbourne, and Sydney regions, for the largest 25 events that occurred between 2014 to 2016. The Melbourne and Sydney regions are similar than other domains in terms of the frequency distribution of hourly precipitation and the discrepancy

between radar and BARRA precipitation.

Overall, the rainfall magnitude corresponding to quantiles is higher for the radar estimates than for BARRA, and this can be attributed to the difference in spatial scale and the area-weighted re-gridding scheme. The difference between two datasets is greater at higher quantiles. However, at around 99% quantiles, the precipitation from BARRA is close to or greater than radar

precipitation.

FSS is calculated for varying quantile thresholds (50, 75, 90, 95, and 99%) at different accumulations across time (1h, 3h and 6h) and presented in Figure 5. As expected, FSS increases with increasing neighbourhood size and decreases with increasing threshold. If there is no frequency bias, then the FSS curve is expected to asymptote to 1 as the neighbourhood size is increased.

In Figure 5, the maximum FSS of 1 is not achieved even at the neighbourhood size of 300 km, which signifies a frequency bias in BARRA.

The FSS scores vary with location, where the regions in increasing order of performance are: Darwin, Brisbane, Melbourne, and Sydney. The results for hourly precipitation and 75% quantile threshold suggest that the skilful spatial scale L

($FSS > 0.5 + p_o/2$) is less than 100 km for all the locations except Darwin. The skilful spatial scale for 90% and 95% quantile thresholds increases to 150 km. For the 99% quantile threshold, BARRA estimates only exhibit useful skill over spatial scales larger than around 250 km.

The FSS metric only provides information on how performance varies with increasing spatial scale. It does not account for

timing errors associated with events that might be initiated at different times and/or evolve at different rates. An indication of such timing errors may be discerned by assessing how the FSS varies with increasing time accumulations. The results shown





in Figure 5 indicate that FSS improves with increasing time accumulations. For example, at Darwin, BARRA is able to provide skilful estimates of 50% threshold of hourly rainfalls only at a scale of 200 km; however, at 3 h and 6 h accumulations, the corresponding spatial scale reduces to 125 km and 100 km, respectively. The accuracy of the BARRA estimates decreases

with increasing rainfall severity, and even at the longest time accumulations, the spatial scale at which rainfalls above the 99% quantiles are skilful extends out to 300 km.

## 5 Discussion

To understand the performance of BARRA precipitation at sub-daily scales, a range of evaluation methods are employed to ascertain its spatial and temporal characteristics. Key insights arising from the results are discussed below.

### 5.1 Performance based on wet day quantiles

The unconditional evaluation of precipitation frequency in terms of wet day quantiles examines the representativeness of sub-daily climatology (Figure 2). The minimum bias across all time accumulations and quantiles in the temperate zone suggests that BARRA provides unbiased estimates of temperate rainfalls over the observed range of events. In addition, an improved performance is observed across all regions when temporal accumulations are considered. This includes the arid and tropical

zones where BARRA performs poorly at an hourly scale. A slight negative bias is observed across all locations in most of the quantiles. This underestimation is however expected due to the mismatch in spatial scale between point observations and grid average precipitation (Maraun, 2013). The point precipitation, in general, is expected to be higher than areal rainfall at 12 km spatial scale.

### 5.2 Spatio-temporal representation of rainfall

A potential displacement of precipitation field in space and time is expected when evaluating high-resolution precipitation datasets, especially when performance is assessed at an hourly timescale and at a point (or single grid cell) location (Rossa et al., 2008). The BARRA estimates exhibit such displacement errors, as evidenced by improved performance at neighbourhood analysis and FSS analysis. An assessment of hourly temporal patterns shows an improvement when neighbourhood grid cells are considered (Figure 3). This suggests that when using precipitation from BARRA, users could benefit from considering

spatial and temporal displacement in the precipitation field, especially during large events. This further suggests an opportunity to utilise hourly distribution of rainfall for disaggregating daily totals.

At the catchment scale, the accuracy of the spatial distribution of rainfall is important for flood modelling. The evaluation of the spatial performance of BARRA against radar data for selected large events showed a mixed result based on locations.

BARRA rainfalls for 90% quantile at 3 h accumulation achieves the target FSS at a spatial scale of range 100-140 km for all





domains except Darwin (250 km) (Figure 5). Achieving useful skill only at a large spatial scale can be partly attributed to the spatial error in the precipitation fields. The higher quantile thresholds are related to small-scale features which are even more likely to be subject to spatial error. Therefore, it is difficult to achieve a skilful spatial scale for these more extreme rainfalls. This has also been pointed out by Roberts and Lean (2008) based on the skill of NWP model outputs at various quantile

thresholds (75, 90, 95 and 99%). The nature of such spatial displacement errors should be considered if BARRA estimates of precipitation are intended to be used for hydrological modelling.

This spatial evaluation is conducted at selected locations to take advantage of the availability of high-resolution blended radar datasets. Across the whole Australian continent there is a dearth of such high-resolution observations of precipitation, and

BARRA could be relied upon to provide estimates of sub-daily precipitation at selected spatial scales in regions where there is a paucity of gauging data.

**5.3 Performance dependence on spatial location**

The overall performance of BARRA varies with location, and this similar trend was evident across all evaluation methods. Both the bias analysis of quantiles and the FSS evaluation show that BARRA performs more poorly in the tropical regions

(Figure 2 and Figure 5). The poorer performance in the tropics, where convective precipitation is dominant, reflects the limitations of the parameterisation scheme to describe sub-grid scale rainfall (Su et al., 2019). This is consistent with the finding of Ebert et al. (2007) that focuses on general performance of numerical weather prediction models. The variation in performance of sub-daily BARRA precipitation across spatial location and different climatologies is consistent with the daily evaluation performed in Acharya et al. (2019).

**5.4 Performance as a function of temporal resolution**

Availability of hourly rainfall observations (based on both pluviometer and radar products) enable BARRA estimates of hourly rainfalls to be evaluated. The uneven and sparse distribution of pluviometer gauges across Australia and the limited availability of radar products is not sufficient for an overall assessment of sub-daily precipitation across the whole of Australia, yet given the wide range of climatologies considered, the analyses provide a comprehensive evaluation of BARRA. Our unconditional

(Figure 2) and direct comparison (Figure 5) both illustrate a similar dependency of performance on temporal resolution. The performance shows a significant improvement when estimates are accumulated from 1h to 3h. The increased performance for coarser temporal resolution can also be linked to the smoothing effect of rainfall at larger accumulation times and reduction of inherent bias.

The trade-off between performance and temporal resolution needs to be considered in combination with the objectives to which the precipitation estimates are used. However, in general terms the increased performance at 3 h and 6 h accumulation suggests





that it is better to use these accumulations from BARRA. Given that the performance of BARRA does vary with location, it is expected that shorter (3 h) accumulations may be appropriate for use in temperate locations and longer accumulations (6 h) in the tropical and arid regions.

## 340 6 Conclusions

An accurate representation of spatial and sub-daily temporal characteristics of precipitation fields is important for many hydro-meteorological applications. BARRA is a regional reanalysis dataset that provides long-term high-resolution estimates of atmospheric variables over the Australian continent. In this study, the spatio-temporal characteristics of sub-daily BARRA precipitation estimates are assessed using various metrics to evaluate its performance at various spatial and temporal scales.
Based on the results, we conclude:

1. Sub-daily precipitation exhibits negative bias at higher quantiles. The magnitude of bias decreases with increasing temporal accumulation and is the lowest at daily accumulations.

2. The spatial displacement of rainfall estimates is evident, especially for rainfall corresponding to higher quantiles.

3. The performance of BARRA varies spatially, and more accurate estimates are provided in southern and eastern 350 Australia (temperate zone) than in northern Australia (tropical zone). These spatial trends in model performance are evident in evaluations undertaken using both gauged point rainfalls and blended radar observations, and are consistent with the evaluation of BARRA estimates of daily rainfall (Acharya et al., 2019).

4. Performance increases with time aggregation, which is expected due to smoothing in accumulated rainfall time series. The performance is reasonably skilful at most of the locations for temporal accumulations of 3h and greater.


A natural interest of the users of BARRA is the potential application of the data in hydro-meteorology. Information on sub-daily rainfalls across the Australian continent is limited due to sparse gauge measurements, the availability of radar at only few locations, spatially coarser global reanalysis products, and few satellite products (~25km, going back to only 1998). BARRA stands out as one of few available datasets at hourly temporal resolution. In that respect, it serves as a useful dataset 360 for the applications requiring sub-daily precipitation. Further, it can be used to help characterise rainfall behaviour in regions where gauges are sparse or non-existent. One of the potential applications of BARRA could be for probabilistic design rainfalls, that is for those engineering applications which utilise information on the relationship between rainfall magnitude and its exceedance probability, either at a point or over an area; such applications are probabilistic in nature and are less sensitive to event-based errors in temporal and spatial behaviour. For applications in flood modelling, where the spatial and temporal 365 distribution of precipitation is important, a proper account needs to be given to the likely spatio-temporal displacement errors in the BARRA precipitation. In addition to direct reliance on absolute estimates of precipitation, it may be that BARRA is well suited to providing information on sub-daily patterns of areal rainfalls that can be scaled to match more reliable estimates of



daily rainfalls. The extent to which such estimates might provide a better understanding of areal sub-daily rainfall behaviour than is possible from gauged point rainfalls warrants further investigation.

**Acknowledgements**

The authors gratefully acknowledge the financial support provided by Seqwater and the Bureau of Meteorology to partially fund SCA's PhD scholarship. We would like to thank Susan Rennie, Kevin Cheong, and Alan Seed (Bureau of Meteorology) for their advice on the use of the radar rainfall products, as retrieved from the Rainfields Archiving System provided by the Bureau of Meteorology. Also, Dörte Jakob and Peter Steinle (Bureau of Meteorology) contributed valuable discussions on general methodology and early results.

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



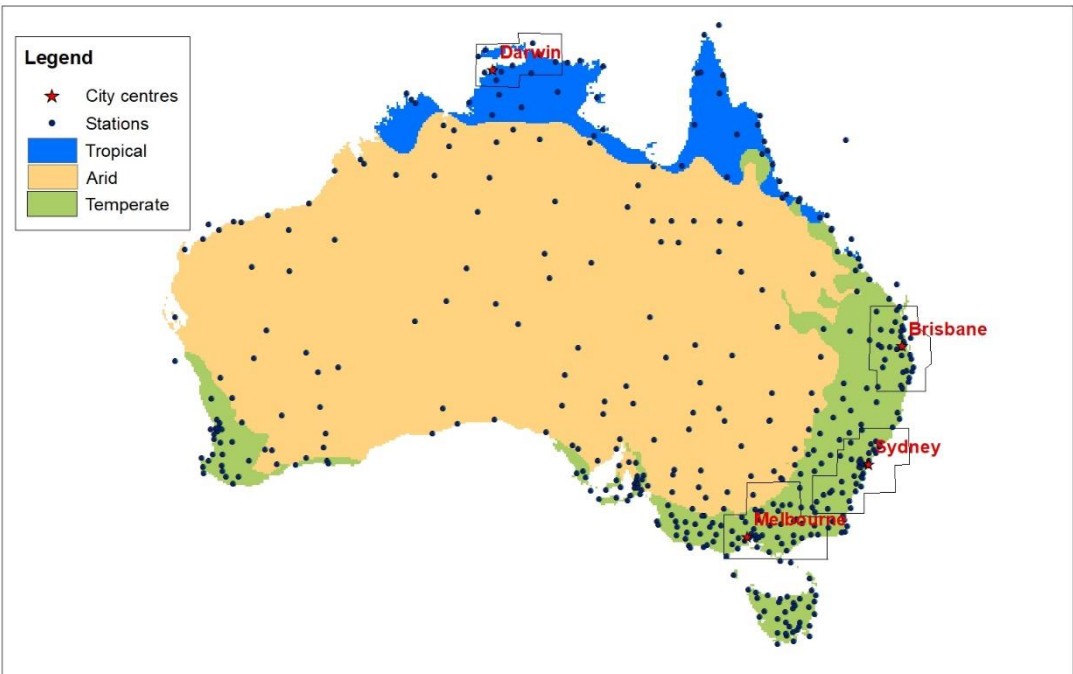


**Figure 1: Study area with locations of pluviometer gauges (points) and the radar data (box surrounding city centres). The locations of radar datasets are Darwin, Brisbane, Sydney and Melbourne regions. The climatic classification is adapted from Peel et al. (2007).**





**Figure 2: Percentage bias in precipitation at quantiles (columns) and accumulations (rows). The boxplots represent the summary of percentage bias across climatic zones for respective rows and columns.**





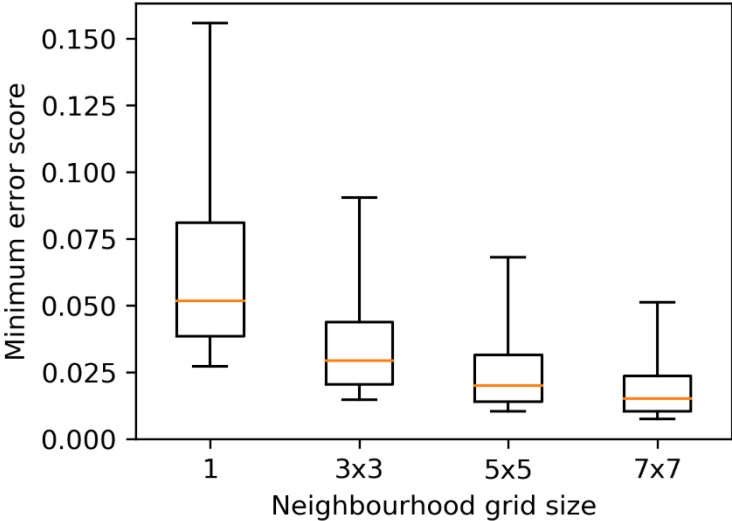

**Figure 3: Boxplots showing the distribution of minimum error score across stations in each neighbourhood grid size. The box represents 25-75th quantile values, horizontal line in the box represents median, and whiskers represent 5-95th quantiles.**





Figure 4: Hourly precipitation rates corresponding to quantiles for BARRA and blended radar data at four different locations.




**Figure 5: Mean FSS as a function of neighbourhood distance for rainfall above quantile thresholds (indicated by colours) at various locations (rows) and accumulations (columns). The dashed horizontal lines indicate the target or uniform (FSS$_{target}$ or FSS$_{uniform}$) skill for each threshold as specified by Roberts and Lean (2008)**