# Peer review of "Ability of an Australian reanalysis dataset to characterise sub-daily precipitation"

_Hydrology and Earth System Sciences, 2019_

## Referee Comment (RC1) · Anonymous Referee #1 · 8 Jan 2020

The manuscript "Ability of an Australian reanalysis dataset to characterize sub-daily precipitation" by Acharya et al. evaluates sub-daily precipitation from a gridded reanalysis dataset, BARRA, covering Australia, against gauge observations and radar data. The aim is to assess the performance of BARRA, specifically for the use in catchment hydrology applications. The aim is clearly stated and the analysis is indeed relevant, as the need for continuous precipitation data of high spatial and temporal resolution is obvious. The manuscript is well structured and has a clear language.

On a general term I have some minor suggestions to improve the manuscript as listed below:

For non-Australians it would be useful to have more information about the climate and topography in Australia, specifically the rainfall climate in the selected areas. Also,

[Figure]

please discuss results in light of regional differences in rainfall climate. I think there is room for one more analysis and figure. Since the target application is catchment hydrology, I would like to see more emphasize on the evaluation of areal precipitation. For instance, a case study analysing the evolution of high-impact events over catchments would be interesting. It might be beneficial to the reader to have other titles on sub-chapters, particularly in the Results section. For instance "4.3 Fractions Skill Score (FSS)" could be renamed to describe what FSS actually evaluates.

Specific comments: Although you state that a spatial resolution of 12 km is considered high, I would argue that the parameterization of convection is a major limitation when studying sub-daily rainfall. Please discuss this in more detail. I can't see that you address the uncertainty in the observation based data. Please discuss these, and perhaps make an attempt to quantify them and include in the figures. p8l228: You state that BARRA tends to overestimate light rain events. Please add a reference or show this in a figure. p10l292: You state that point precipitation is generally higher than areal rainfall at 12 km scale. Could you suggest a way to overcome this? Could you consider using an areal reduction factor? If not, why? In many studies lately there has been a focus on quantifying the contribution of changes in intensity and changes in the frequency to trends in (heavy) precipitation. Could you please relate your results to the how well BARRA represents intensity and frequency?

Figures: Figure 1: Please thicken the line marking the four study regions. Figure 2: Although I like this figure, it is a bit hard to see the colors etc due to the small maps. Could you split the maps and the boxplots into two figures? Figure 5: I might have missed something, but I do not understand why you here only study precipitation up to 6 hours, while up to 24 hours in Figure 2.

Technical corrections: p2l32-33: This sentence should be rephrased. p2l38: Remove "a". p2l52: "developing use cases"? Please rephrase. p9l258: Add "the" before "two datasets" p10l304: Do you mean "mixed result between locations"?

---

## Referee Comment (RC2) · Anonymous Referee #2 · 10 Feb 2020

The article is sound and makes a modest contribution (by examining sub-daily time scales) to body of literature on the evaluation of reanalysis rainfall data. I think it should be published subject to some clarification. Some items for correction / clarification:

L19: quantile should be replaced with percentile.

L47-48: what does it mean that BARRA is driven by ERA-Interim?

L104 - 115: this seems an overly critical view of radar rainfall. IN the assessment of the sources of error in radar retrievals, are the authors referring to their own gauge-correction procedure or other published work? Further, I understand that the authors use the radar data to evalute the spatial distribtion of rainfall, but how does aggregating the 1.5km data to 12km (which I assume they did for a fair evaluation) change the interpretation of the spatial patterns for such small regions of Australia. At 12km resolution, an evaluation against satellite retrievals (e.g. GPM IMERG final product) may provide the same information but for the whole country.

* Box plots in Fig. 2 are very difficult to see on the printed verison. Perhaps a landscape layout for figure 2 might help?

---

## Author Comment (AC1) · 11 Mar 2020

**Response to reviews on the manuscript hess-2019-432 "Ability of an Australian reanalysis dataset to characterise sub-daily precipitation" by Suwash Chandra Acharya et al.**

We would like to thank the two anonymous referees for their constructive comments and suggestions on our paper.

In the following sections we provide a detailed response to all the remarks and suggestions made by the referees to improve the manuscript. The reviewers' comments (in black), our corresponding reply (in blue), and proposed modifications (underlined).

**Response to Referee #1**

**General comments**

The manuscript "Ability of an Australian reanalysis dataset to characterize sub-daily precipitation" by Acharya et al. evaluates sub-daily precipitation from a gridded reanalysis dataset, BARRA, covering Australia, against gauge observations and radar data. The aim is to assess the performance of BARRA, specifically for the use in catchment hydrology applications. The aim is clearly stated and the analysis is indeed relevant, as the need for continuous precipitation data of high spatial and temporal resolution is obvious. The manuscript is well structured and has a clear language.

We thank Referee #1 for acknowledging the relevance of the analysis presented and overall positive remarks on the manuscript.

On a general term I have some minor suggestions to improve the manuscript as listed below:

For non-Australians it would be useful to have more information about the climate and topography in Australia, specifically the rainfall climate in the selected areas. Also, please discuss results in light of regional differences in rainfall climate. I think there is room for one more analysis and figure.

We agree that including a description of Australian climate and topography will help non-Australian readers to generalise the conclusions of the paper for other studies. In Figure 1, we present the climate classification across Australia, based on the Köppen-Geiger classification which is not adequately discussed within the text. To address this comment, we will revisit the section "2. Study and data sources" to describe the topography and climate zones across Australia and the selected areas.

In the manuscript subsection "5.2 Performance dependence on spatial location", we discuss the results focusing on the location of selected areas. This discussion implicitly considers the differences in climate of the selected areas. However, after describing the climate classification and rainfall climate of the selected areas, we will further extend the discussion to explicitly present the variation in results across the different climatic zones and rainfall climate.

Since the target application is catchment hydrology, I would like to see more emphasize on the evaluation of areal precipitation. For instance, a case study analysing the evolution of high-impact events over catchments would be interesting.

We also recognise the importance of areal precipitation in the catchment hydrological applications. The evaluation and use of areal rainfalls require a substantial assessment in itself, and indeed we are in the final stages of preparing a paper that explores the evaluation and design applications of areal rainfall estimates. We will add a comment on the need for further assessment of areal precipitation of the datasets in the Discussion or Conclusion section.

It might be beneficial to the reader to have other titles on sub-chapters, particularly in the Results section. For instance, "4.3 Fractions Skill Score (FSS)" could be renamed to describe what FSS actually evaluates.

We will re-name the sub-chapters in the methodology and results chapters to be more descriptive.

For example, in Methodology and corresponding Results section:

3.1 "Frequency distribution of rainfall" to "Frequency distribution of sub-daily rainfall"

3.2 "Spatio-temporal distribution" to "Neighbourhood-based diurnal patterns"

3.3 "Fractions skill score (FSS)" to "Neighbourhood-based spatial evaluation"

**Specific comments**

Although you state that a spatial resolution of 12 km is considered high, I would argue that the parameterization of convection is a major limitation when studying sub-daily rainfall. Please discuss this in more detail.

Our statement that the spatial resolution of BARRA is high is based on its comparison to other available reanalysis datasets for Australia. However, we agree that the spatial resolution is not fine enough for parameterisation of convection.

The Unified Model's parameterised sub-grid convection scheme (the forecast model used in BARRA) which works independently at each grid point, produces a bias towards widespread precipitation (Clark et al., 2016; Su et al., 2019). This parameterisation scheme (for detail, see: Su et al., 2019) adopted for sub-grid convection is limiting in resolving convective rainfall and affects the locations dominated by such rainfall (especially tropics). This is observed in daily evaluation of BARRA by (Acharya et al., 2019) where the performance was better in temperate than tropical regions. We will further discuss the implication of convection parameterisation in BARRA in relation to analysing sub-daily rainfall in datasets and/or discussion section of the manuscript.

I can't see that you address the uncertainty in the observation based data. Please discuss these, and perhaps make an attempt to quantify them and include in the figures.

In our study, we use two datasets as a benchmark for evaluation: gauge and blended radar. In absence of any other alternative of high-quality datasets, these two datasets represent the best available point and spatial estimates of rainfall and thus provide an appropriate basis for comparison. We acknowledge

the uncertainty arising due to comparison of point rainfall against BARRA grid and have discussed it accordingly. However, without suitable reference data sets it is not possible to calculate the uncertainty in these sub-daily observations and there is no published study on this which is relevant to the regions we studied. Similarly, the radar datasets are prone to various error sources which are discussed in the "2. Study area and data sources" section of the manuscript. The Bureau of Meteorology have blended radar estimates with gauged data and have estimated the associated uncertainties however, this information has not been published and is not publicly available.

p8l228: You state that BARRA tends to overestimate light rain events. Please add a reference or show this in a figure.

Overestimation of light rainfall at daily scale is documented in (Acharya et al., 2019; Su et al., 2019). We will add reference/s to the statement in the revised manuscript.

p10l292: You state that point precipitation is generally higher than areal rainfall at 12 km scale. Could you suggest a way to overcome this? Could you consider using an areal reduction factor? If not, why? In many studies lately there has been a focus on quantifying the contribution of changes in intensity and changes in the frequency to trends in (heavy) precipitation. Could you please relate your results to the how well BARRA represents intensity and frequency?

Despite an apparent mismatch in spatial resolution/representation, we use point rainfall as a benchmark for evaluating BARRA rainfall because it is one of the best available datasets for evaluation. Accordingly, we discuss the variation in performance in light of the difference in spatial scale. Addressing these differences arising due to varying representativity of point and areal rainfall is not straightforward. In "design rainfall" related applications, areal reduction factors are applied to scale the point rainfall to areal rainfall, however, we note that 1) such factors could lack the actual properties of large rainfall events and result in mis-estimation of flood risks (Wright et al., 2014), and 2) there is limited robust evidence for ARF factors at this small scale (Podger et al., 2015; Stensmyr et al., 2015). In hydrological modelling applications, any constant scaling applied to point rainfall would not hold true across entire time series due to spatial variability of rainfall datasets. One way of addressing such differences could be to evaluate BARRA rainfall against high-quality reference spatial datasets, but such data sets would require considerable effort to derive.

The focus of our study is to present an assessment of sub-daily rainfall from BARRA at point and spatial scale. The period of evaluation is limited to a six-year period (2010-2015) based on the availability of observed (benchmark) datasets. Any comment on the change in intensity, frequency and associated trends in rainfall is limited due to the temporal extent of analysis in the study and is beyond the scope of the paper. However, we encourage the future assessments of trends in intensities and frequencies of heavy rainfall based on BARRA rainfall dataset.

**Figures**
Figure 1: Please thicken the line marking the four study regions.

We will edit the figure accordingly in the revised manuscript.

Figure 2: Although I like this figure, it is a bit hard to see the colors etc due to the small maps. Could you split the maps and the boxplots into two figures?

We agree that the figure as currently presented is slightly difficult to read. We will present an improved plot either by splitting the maps and the boxplots, or by changing the orientation of the plot in the revised manuscript.

Figure 5: I might have missed something, but I do not understand why you here only study precipitation up to 6 hours, while up to 24 hours in Figure 2.

Figure 2 and Figure 5 are results from two different evaluation approaches: non-conditional frequency distribution at a point, and rainfall events over an area. For the former, we explore the bias in rainfall intensity at various frequencies and temporal accumulations up to 24 hours. Our attempt to compute the frequencies up to 24 hours was to understand the variation in rainfall frequencies at different temporal accumulations. As discussed in introduction section, this assessment could be useful for developing intensity-frequency curves for design applications.

With the areal analysis, we use Fractions Skill Scores (FSS) to understand the representativeness of spatial patterns of rainfall from BARRA at sub-daily scales. Our analysis of temporal accumulations of 3 and 6 hours was undertaken to assess the utility of the areal rainfalls at sub-daily temporal scales that are relevant to hydrological modelling. The evaluation of multiple sub-daily aggregations (1, 3 and 6 hours) allows us to determine a suitable temporal aggregation for hydrological modelling purposes. While extending this accumulation to 24 hours will definitely show improved metrics, the resulting time series would have more limited application to hydrological modelling.

In response to this comment, we will clarify the choice of temporal accumulations and their utility in the Methodology section.

**Technical corrections:**
p2l32-33: This sentence should be rephrased.

p2l38: Remove "a".

p2l52: "developing use cases"? Please rephrase.

p9l258: Add "the" before "two datasets"

p10l304: Do you mean "mixed result between locations"?

We will address these technical corrections in the revised manuscript.

**Response to Referee #2**

**General comments**

The article is sound and makes a modest contribution (by examining sub-daily time scales) to body of literature on the evaluation of reanalysis rainfall data. I think it should be published subject to some clarification.

We thank Referee #2 for their positive remarks on the paper, and for their constructive comments for improving the manuscript.

**Specific comments**

Some items for correction / clarification:

L19: quantile should be replaced with percentile.

Agreed. We will make corrections in the revised manuscript.

L47-48: what does it mean that BARRA is driven by ERA-Interim?

The initial and boundary conditions required for BARRA is obtained from ERA-Interim. We will edit the sentence to make it clearer.

L104 - 115: this seems an overly critical view of radar rainfall. In the assessment of the sources of error in radar retrievals, are the authors referring to their own gauge correction procedure or other published work? Further, I understand that the authors use the radar data to evaluate the spatial distribution of rainfall, but how does aggregating the 1.5km data to 12km (which I assume they did for a fair evaluation) change the interpretation of the spatial patterns for such small regions of Australia. At 12km resolution, an evaluation against satellite retrievals (e.g. GPM IMERG final product) may provide the same information but for the whole country.

Our assessment of radar was based on the review of various studies mentioned in the paper. The blended radar data were made available from Bureau of Meteorology, Australia. The blended radar had already gone through the gauge correction procedures and any assessment regarding such corrections is beyond the scope of the paper.

We properly acknowledge that the radar dataset is the best available spatial dataset and provides an accurate estimate of spatio-temporal distribution of rainfall. In addition, we apply area-weighted approach to re-grid BARRA (~12km) to radar grid (~1.5). It can be expected that the re-gridding of BARRA will underestimate the intensity of rainfall at a finer scale. To address this, we apply percentile-based threshold while calculating Fractions Skill Score (FSS) to evaluate spatial distribution of rainfall field. Nonetheless, any results from FSS obtained for a spatial scale less than <12km should be interpreted carefully.

Similarly, we agree with the concept that evaluating the BARRA across the whole country would provide valuable information. Such evaluation, however, would be limited by the availability of high-quality and high-resolution benchmark datasets. As mentioned in the comment by Reviewer #2, satellite retrievals such as the IMERG final product could possibly be used for such a large-scale evaluation. Assessments of the IMERG final product have been shown to perform better than TRMM or the IMERG initial run (Beck et al., 2019; Wang et al., 2017), however, a majority of such evaluations

are limited to daily scales (Beck et al., 2019; Wang and Yong, 2020). As our assessment focuses on sub-daily rainfall from the novel regional reanalysis dataset (BARRA) we would expect the benchmark data to be accurate at that temporal scale. Currently, there are no comprehensive assessment of the IMERG final run at sub-daily scales for Australian continent, and this limits our study to evaluate against a more accurate radar datasets at selected locations. A more detailed spatial assessment of BARRA would be possible once further comprehensive assessments of high-resolution satellite datasets are available.

In the revised manuscript, we will clarify the rationale for the choice of study areas and reference datasets used in the current study. We will further discuss our current limitations in evaluating over entire Australia and provide comment on possible directions for future assessments.

Box plots in Fig. 2 are very difficult to see on the printed version. Perhaps a landscape layout for figure 2 might help?

We agree that the figure, currently, is difficult to read. We will present an improved plot either by splitting the maps and the boxplots, or changing the orientation of the plot in the revised manuscript.

**References**

Acharya, S. C., Nathan, R., Wang, Q. J., Su, C.-H. and Eizenberg, N.: An evaluation of daily precipitation from a regional atmospheric reanalysis over Australia, Hydrol. Earth Syst. Sci., 23(8), 3387–3403, doi:10.5194/hess-23-3387-2019, 2019.

Beck, H. E., Pan, M., Roy, T., Weedon, G. P., Pappenberger, F., van Dijk, A. I. J. M., Huffman, G. J., Adler, R. F. and Wood, E. F.: Daily evaluation of 26 precipitation datasets using Stage-IV gauge-radar data for the CONUS, Hydrol. Earth Syst. Sci., 23(1), 207–224, doi:10.5194/hess-23-207-2019, 2019.

Clark, P., Roberts, N., Lean, H., Ballard, S. P. and Charlton-Perez, C.: Convection-permitting models: A step-change in rainfall forecasting, Meteorol. Appl., 23(2), 165–181, doi:10.1002/met.1538, 2016.

Ebert, E. E., Janowiak, J. E. and Kidd, C.: Comparison of near-real-time precipitation estimates from satellite observations and numerical models, Bull. Am. Meteorol. Soc., 88(1), 47–64, doi:10.1175/BAMS-88-1-47, 2007.

de Leeuw, J., Methven, J. and Blackburn, M.: Evaluation of ERA-Interim reanalysis precipitation products using England and Wales observations, Q. J. R. Meteorol. Soc., 141(688), 798–806, doi:10.1002/qj.2395, 2015.

Podger, S., Green, J., Jolly, C., Beesley, C. and others: Creating long duration areal reduction factors, in 36th Hydrology and Water Resources Symposium: The art and science of water, p. 39, Engineers Australia., 2015.

Stensmyr, P., Babister, M., Adam, M. and others: Short duration areal reduction factors: Sydney, Melbourne and Brisbane, in 36th Hydrology and Water Resources Symposium: The art and science of water, p. 121., 2015.

Su, C.-H., Eizenberg, N., Steinle, P., Jakob, D., Fox-Hughes, P., White, C. J., Rennie, S., Franklin, C., Dharssi, I. and Zhu, H.: BARRA v1.0: the Bureau of Meteorology Atmospheric high-resolution Regional Reanalysis for Australia, Geosci. Model Dev., 12(5), 2049–2068, doi:10.5194/gmd-12-2049-2019, 2019.

Wang, H. and Yong, B.: Quasi-Global Evaluation of IMERG and GSMaP Precipitation Products over

Land Using Gauge Observations, Water, 12(1), 243, doi:10.3390/w12010243, 2020.

Wang, Z., Zhong, R., Lai, C. and Chen, J.: Evaluation of the GPM IMERG satellite-based precipitation products and the hydrological utility, Atmos. Res., 196, 151–163, doi:10.1016/j.atmosres.2017.06.020, 2017.

Wright, D. B., Smith, J. A. and Baeck, M. L.: Critical Examination of Area Reduction Factors, J. Hydrol. Eng., 19(4), 769–776, doi:10.1061/(ASCE)HE.1943-5584.0000855, 2014.

---

## Author Response (AR1)

**Response to reviews on the manuscript hess-2019-432 "Ability of an Australian reanalysis dataset to characterise sub-daily precipitation" by Suwash Chandra Acharya et al.**

We would like to thank the two anonymous referees for their constructive comments and suggestions on our paper.

In the following sections we provide a detailed response to all the remarks and suggestions made by the referees to improve the manuscript. The reviewers' comments (in black), our corresponding reply (in blue), and revisions (*italicised*).

**Response to Referee #1**

**General comments**

The manuscript "Ability of an Australian reanalysis dataset to characterize sub-daily precipitation" by Acharya et al. evaluates sub-daily precipitation from a gridded reanalysis dataset, BARRA, covering Australia, against gauge observations and radar data. The aim is to assess the performance of BARRA, specifically for the use in catchment hydrology applications. The aim is clearly stated and the analysis is indeed relevant, as the need for continuous precipitation data of high spatial and temporal resolution is obvious. The manuscript is well structured and has a clear language.

We thank Referee #1 for acknowledging the relevance of the analyses presented and overall positive remarks on the manuscript.

On a general term I have some minor suggestions to improve the manuscript as listed below:

For non-Australians it would be useful to have more information about the climate and topography in Australia, specifically the rainfall climate in the selected areas. Also, please discuss results in light of regional differences in rainfall climate. I think there is room for one more analysis and figure.

We agree that including a description of Australian climate and topography will help non-Australian readers to generalise the conclusions of the paper for other studies. In Figure 1, we present the climate classification across Australia, based on the Köppen-Geiger classification which is not adequately discussed within the text. To address this comment, we have revised the study area and data sources section (Lines: 95-102) as follows:

*The pluviometer observations in this study encompass three of the broad climatic zones across Australia, namely tropical, temperate and arid (**Error! Reference source not found.**). Rainfall in Australia is highly variable across space and time. Rainfall is concentrated during summer in the tropical north, whereas rainfall is more prevalent during winter in the temperate south and southwest. The southeast region experiences a more consistent rainfall throughout the year. The central arid region receives the least total rainfall, and the eastern coast the highest average rainfall. The spatial evaluation across the entire Australia is impeded by the availability of high-resolution observation-based dataset. Therefore, the spatial evaluation is undertaken by using radar-based datasets at four*

*city-centred regions (Brisbane, Darwin, Melbourne and Sydney). These study areas are located in the tropical and temperate zones with varying rainfall climates.*

In the manuscript subsection "5.3 Performance dependence on spatial location", we discuss the results focusing on the location of selected areas. This discussion implicitly considers the differences in climate of the selected areas. However, after describing the climate classification and rainfall climate of the selected areas, we have revised the discussion section (Lines: 329-332) as follows:.

*Both the bias analysis of quantiles and the FSS evaluation show that the performance of BARRA gradually improves as we move from northern to southern regions (**Error! Reference source not found.** and **Error! Reference source not found.**). This variation in performance can partly be attributed to the different rainfall climate at these regions. The convective precipitation during summer season is dominant at northern (low) latitudes, whereas winter (or uniform) non-convective rainfall are dominant at southern (high) latitudes.*

Since the target application is catchment hydrology, I would like to see more emphasize on the evaluation of areal precipitation. For instance, a case study analysing the evolution of high-impact events over catchments would be interesting.

We also recognise the importance of areal precipitation in the catchment hydrological applications. The evaluation and use of areal rainfalls require a substantial assessment in itself, and indeed we are in the final stages of preparing a paper that explores the evaluation and design applications of areal rainfall estimates. We have revised the conclusion section (Lines: 384, 388) as follows:.

*The ability of BARRA to provide areal rainfall estimates is useful in catchment hydrological applications.*

*…*

*Accordingly, future work will be directed towards evaluation and design applications of areal rainfall estimates.*

It might be beneficial to the reader to have other titles on sub-chapters, particularly in the Results section. For instance, "4.3 Fractions Skill Score (FSS)" could be renamed to describe what FSS actually evaluates.

We have re-named the sub-chapters to be more descriptive. The sub-chapters in the Methodology and corresponding Results section are revised as follows:

*3.1 "Frequency distribution of rainfall" to "Frequency distribution of sub-daily rainfall"*

*3.2 "Spatio-temporal distribution" to "Neighbourhood-based diurnal patterns at point locations"*

*3.3 "Fractions skill score (FSS)" to "Neighbourhood-based spatial evaluation"*

**Specific comments**

Although you state that a spatial resolution of 12 km is considered high, I would argue that the parameterization of convection is a major limitation when studying sub-daily rainfall. Please discuss this in more detail.

Our statement that the spatial resolution of BARRA is "high" is based on its comparison to other available reanalysis datasets for Australia. However, we agree that the spatial resolution is not fine enough for parameterisation of convection.

The Unified Model's parameterised sub-grid convection scheme (the NWP model used in BARRA) which works independently at each grid point, produces a bias towards widespread precipitation (Clark et al., 2016; Su et al., 2019). The parameterisation scheme (for detail, see: Su et al., 2019) adopted for sub-grid convection is limiting in resolving convective rainfall and affects the locations dominated by such rainfall (especially tropics). This is observed in daily evaluation of BARRA by (Acharya et al., 2019) where the performance was better in temperate than tropical regions. We have revised the study area and data sources section (Lines: 137-138), and discussion section (Lines: 329-332) as follows:

*This parameterisation scheme adopted for sub-grid convection is limiting in resolving convective rainfall and affects the locations dominated by such rainfall (especially tropics).*

*…*

*Both the bias analysis of quantiles and the FSS evaluation show that the performance of BARRA gradually improves as we move from northern to southern regions (**Error! Reference source not found.** and **Error! Reference source not found.**). This variation in performance can partly be attributed to the different rainfall climate at these regions. The convective precipitation during summer season is dominant at northern (low) latitudes, whereas winter (or uniform) non-convective rainfall are dominant at southern (high) latitudes.*

I can't see that you address the uncertainty in the observation based data. Please discuss these, and perhaps make an attempt to quantify them and include in the figures.

In our study we use two datasets as a benchmark for evaluation: gauge and blended radar. In absence of any other alternative of high-quality datasets, these two datasets represent the best available point and spatial estimates of rainfall and thus provide an appropriate basis for comparison. We acknowledge the uncertainty arising due to comparison of point rainfall against BARRA grid and have discussed it accordingly. However, without suitable reference data sets it is not possible to calculate the uncertainty in these sub-daily observations and there is no published study on this which is relevant to the regions we studied. Similarly, the radar datasets are prone to various error sources which are discussed in the "2. Study area and data sources" section of the manuscript. The Bureau of Meteorology have blended radar estimates with gauged data and have estimated the associated uncertainties however, this information has not been published and is not publicly available.

p8l228: You state that BARRA tends to overestimate light rain events. Please add a reference or show this in a figure.

A reference has been added and the sentence reads (Lines:240-241 ) *This difference is partly due to the tendency of BARRA to overestimate light rain events (Su et al., 2019).*

p10l292: You state that point precipitation is generally higher than areal rainfall at 12 km scale. Could you suggest a way to overcome this? Could you consider using an areal reduction factor? If not, why? In many studies lately there has been a focus on quantifying the contribution of changes in intensity and changes in the frequency to trends in (heavy) precipitation. Could you please relate your results to the how well BARRA represents intensity and frequency?

Despite an apparent mismatch in spatial resolution/representation, we use point rainfall as a benchmark for evaluating BARRA rainfall because it is one of the best available datasets for evaluation. Accordingly, we discuss the variation in performance in light of the difference in spatial scale. Addressing the performance variation arising from differences in the spatial representation of point and areal rainfalls is not straightforward. In "design rainfall" applications (ie those based on probabilistic representations of rainfall intensities for design purposes), areal reduction factors are applied to scale point rainfalls to areal rainfalls, however, we note that 1) such factors could lack the actual properties of large rainfall events and result in mis-estimation of flood risks (Wright et al., 2014), and 2) there is limited robust evidence for areal reduction factors at this small scale (Podger et al., 2015; Stensmyr et al., 2015). In hydrological modelling applications, any constant scaling applied to point rainfall would not hold true for individual events as they are derived to preserve statistical characteristics of the entire time series. One way of addressing such differences could be to evaluate BARRA rainfall against high-quality reference spatial datasets, but such data sets would require considerable effort to derive.

The focus of our study is to present an assessment of sub-daily rainfall from BARRA at point and spatial scales, where the period of evaluation is limited to a six-year period (2010-2015) based on the availability of observed (benchmark) datasets. Given the limited time period of the BARRA data available at the time of this study, it is difficult to make any meaningful inferences regarding trends in the intensity and frequency of rainfalls. However, it is hoped that future assessments will consider the contribution of changes in rainfall intensity and frequency to trends in (heavy) precipitation.

**Figures**

Figure 1: Please thicken the line marking the four study regions.

Addressed in the revised manuscript.

Figure 2: Although I like this figure, it is a bit hard to see the colors etc due to the small maps. Could you split the maps and the boxplots into two figures?

We agree that the figure as currently presented is slightly difficult to read. Since the maps and boxplots are linked together, following reviewer #2' suggestion, we have changed the orientation of the plot to make it bigger in the revised manuscript.

Figure 5: I might have missed something, but I do not understand why you here only study precipitation up to 6 hours, while up to 24 hours in Figure 2.

Figure 2 and Figure 5 are results from two different evaluation approaches: non-conditional frequency distribution at a point, and rainfall events over an area. For the former, we explore the bias in rainfall

intensity at various frequencies and temporal accumulations up to 24 hours. Our attempt to compute the frequencies up to 24 hours was to understand the variation in rainfall frequencies at different temporal accumulations. As discussed in introduction section, this assessment could be useful for developing intensity-frequency curves for design applications.

With the areal analysis, we use Fractions Skill Scores (FSS) to understand the representativeness of spatial patterns of rainfall from BARRA at sub-daily scales. Our analysis of temporal accumulations of 3 and 6 hours was undertaken to assess the utility of the areal rainfalls at sub-daily temporal scales that are relevant to hydrological modelling. The evaluation of multiple sub-daily aggregations (1, 3 and 6 hours) allows us to determine a suitable temporal aggregation for hydrological modelling purposes. While extending this accumulation to 24 hours will definitely show improved metrics, the resulting time series would have more limited application to hydrological modelling.

In response to this comment, we have revised the Methodology section (Lines: 157-158, 229-231) as follows:

*The analysis of frequencies corresponding to different accumulations (up to 24 h) is selected to be of relevance to design rainfall applications.*

*...*

*It is worth noting that FSS metric only provides information on variation of performance with increasing spatial scale. The timing errors at finer temporal scales can be indirectly discerned by analysing variation of FSS with time accumulations. Therefore, the FSS is evaluated at temporal aggregations of 3 h and 6 h.*

**Technical corrections:**
p2l32-33: This sentence should be rephrased. p2l38: Remove "a".

p2l52: "developing use cases"? Please rephrase.

p9l258: Add "the" before "two datasets"

p10l304: Do you mean "mixed result between locations"?

These technical corrections have been addressed in the revised manuscript.

**Response to Referee #2**

**General comments**
The article is sound and makes a modest contribution (by examining sub-daily time scales) to body of literature on the evaluation of reanalysis rainfall data. I think it should be published subject to some clarification.

We thank Referee #2 for their positive remarks on the paper, and for their constructive comments for improving the manuscript.

**Specific comments**

Some items for correction / clarification:

L19: quantile should be replaced with percentile.

Agreed. We have made corrections in the revised manuscript.

L47-48: what does it mean that BARRA is driven by ERA-Interim?

The initial and boundary conditions required for BARRA is obtained from ERA-Interim. We have revised the sentence (Lines: 46-49) to make it clearer.

*It is driven with initial and boundary conditions from the global ERA-Interim reanalysis (~79 km) and provides estimates over the Australasian region from 1990 to 2018.*

L104 - 115: this seems an overly critical view of radar rainfall. In the assessment of the sources of error in radar retrievals, are the authors referring to their own gauge correction procedure or other published work? Further, I understand that the authors use the radar data to evaluate the spatial distribution of rainfall, but how does aggregating the 1.5km data to 12km (which I assume they did for a fair evaluation) change the interpretation of the spatial patterns for such small regions of Australia. At 12km resolution, an evaluation against satellite retrievals (e.g. GPM IMERG final product) may provide the same information but for the whole country.

Our assessment of radar was based on the review of various studies mentioned in the paper. The blended radar data were made available from Bureau of Meteorology, Australia. The blended radar data had already undergone gauge correction procedures and any assessment regarding such corrections is beyond the scope of the paper.

We acknowledge that the radar dataset is the best available spatial dataset and provides an accurate estimate of spatio-temporal distribution of rainfall. In addition, we apply area-weighted approach to re-grid BARRA (~12km) to radar grid (~1.5). It can be expected that the re-gridding of BARRA will underestimate the intensity of rainfall at a finer scale. To address this, we apply a percentile-based threshold while calculating Fractions Skill Score (FSS) to evaluate the spatial distribution of the rainfall field. Nonetheless, any results from FSS obtained for a spatial scale less than <12km needs to be interpreted carefully.

Similarly, we agree that evaluating BARRA across the whole country would provide valuable information. Such evaluation, however, would be limited by the availability of high-quality and high-resolution benchmark datasets. As mentioned in the comment by Reviewer #2, satellite retrievals such as the IMERG final product could possibly be used for such a large-scale evaluation. Assessments of the IMERG final product have been shown to perform better than TRMM or the IMERG initial run (Beck et al., 2019; Wang et al., 2017), however, a majority of such evaluations are limited to daily scales (Beck et al., 2019; Wang and Yong, 2020). As our assessment focuses on sub-daily rainfall from the BARRA reanalysis dataset it is necessary to use benchmark data that is accurate at that temporal scale. However, there are currently no comprehensive assessments of the IMERG final run at sub-daily scales for Australian continent, and this forces us to focus our evaluation on more accurate radar datasets at selected locations. A more detailed spatial assessment of BARRA would be possible once further comprehensive assessments of high-resolution satellite datasets are available.

In the revised manuscript, we have clarified the rationale for the choice of study areas and reference datasets used in the current study. We have further discussed our current limitations in evaluating over entire Australia and provided comment on possible directions for future assessments. The manuscript is revised as follows (Lines: 99-101, 367-371):

*The spatial evaluation across the entire Australia is impeded by the availability of high-resolution observation-based datasets. Therefore, the spatial evaluation is undertaken by using high quality radar-based datasets at four city-centred regions (Brisbane, Darwin, Melbourne and Sydney).*

*…*

*One of the limitations of this study is that the spatial assessment of BARRA is restricted to a few locations where radar-based datasets are available and the performance of BARRA across Australia is generalised based on evaluation at those locations. Currently, the lack of high-quality and high-resolution benchmark datasets limits our ability to fully understand the performance of BARRA across the entire continent. However, a more detailed spatial assessment of BARRA would be possible once such benchmark datasets are available.*

Box plots in Fig. 2 are very difficult to see on the printed version. Perhaps a landscape layout for figure 2 might help?

We agree that the figure, currently, is difficult to read. We have changed the orientation of the plot in the revised manuscript.

[revised manuscript text omitted]

---

## Author Response (AR2)

**Response to Editor's comment on the manuscript hess-2019-432 "Ability of an Australian reanalysis dataset to characterise sub-daily precipitation" by Suwash Chandra Acharya et al.**

We would like to thank the editor Micha Werner for his very constructive comments and suggestions on our paper. The specificity of these comments and the evident care given to reading the paper is greatly appreciated. In the following, we have grouped the comments and provided a response to the comments and suggestions by the editor to improve the manuscript. The editor's comments (in black), our corresponding reply (in blue), and modifications (underlined).

**Technical corrections**

L54 Please check this sentence - appears as if there is a word missing (large precipitation amounts)

L99-100 Rephrase sentence: The spatial evaluation across the whole of Australia is impeded by the lack of the availability of a high-resolution observation-based dataset.

L331 climate in these regions

L 332 would it be more suitable to call this "frontal" rainfall?

L 378 This sentence should be revised. A suggestion: One of the potential applications of BARRA is for deriving probabilistic design rainfall events for engineering applications, which utilise information on the relationship between rainfall magnitude and its exceedance probability either at a point or over an area. Such applications are probabilistic in nature and are less sensitive to spatial and temporal errors of the individual events.

We agree to these suggestions made and we have revised the sentences accordingly.

L164 while I appreciate the change of title, I do not think diurnal is the appropriate word as this suggest a day/night pattern which is not implied. Also change in the results section.

We have replaced "diurnal" with "sub-daily" which is appropriate for this study. The revised the title of sub-sections in methodology (3.2) and results section (4.2) to "Neighbourhood-based sub-daily patterns at point locations".

**Conclusion: Summary points**

L 358-359 This conclusion should be made a little more specific. First indicate that you mean the Sub-daily precipitation estimates from BARRA. That is somewhat implicit. Also check the second sentence. The statement that it is lowest at daily accumulation I assume only comes from the fact that daily is the largest temporal aggregation considered. I would assume the bias will continue to decrease as the temporal accumulation increases beyond the day.

L 360 As with the previous conclusion this sentence needs to be rephrased as it does not in itself represent a conclusion. This should be self-standing, and not suggest that it is evident. Please rephrase.

L363-364 While this conclusion is well put (and clearer than the previous two, it is not common to include references in the conclusions. That is reserved for the discussion. I would suggest to drop the phrase after the comma, and double check that this is addressed in the discussion (which I believe it is).

L 365 This seems very similar to the second conclusions. So perhaps in strengthening that as suggested these can be combined.

We agree with the editor's suggestions to make the conclusion more specific. We have explicitly included "BARRA" where deemed necessary. We have combined the points summarising the effect of temporal accumulations. The revised conclusion reads as follows:

1. Sub-daily precipitation from BARRA exhibits negative bias at higher quantiles. The magnitude of bias varies with event severity and temporal accumulation.

2. There is some tendency for BARRA reanalysis rainfalls to exhibit spatial displacement, and this is more pronounced for rainfall corresponding to higher quantiles.

3. The performance of BARRA precipitation depends on spatial location with poorer performance in tropical relative to temperate regions. These spatial trends are consistent across evaluations undertaken using both gauged point rainfalls and blended radar observations.

4. Bias in BARRA precipitation quantiles at point scale and spatial displacement errors at spatial scale decrease with increasing time aggregation and the performance is reasonably skilful at most of the locations for temporal accumulations of 3h and greater.

**Conclusion: Final paragraph**

L384-389 I find this final paragraph a little weak and seems to suggest work that should be done is the work that is presented. First I would reconsider using words as "may" in the final paragraph of the conclusions as it leaves the reader somewhat unsure. Also, I would assume that further research should be directed at the behaviour of hydrological fluxes at sub-daily scale using sub-daily precipitation from BARRA. I also suggest to look again at the last sentence as I am not so sure if design applications will address what the sentences before suggests.

We agree with the point raised by the editor. Accordingly, we have revised the final paragraph as follows:

[revised manuscript text omitted]